# Ventilator Weaning in Prolonged Mechanical Ventilation—A Narrative Review

**DOI:** 10.3390/jcm13071909

**Published:** 2024-03-26

**Authors:** Tamás Dolinay, Lillian Hsu, Abigail Maller, Brandon Corbett Walsh, Attila Szűcs, Jih-Shuin Jerng, Dale Jun

**Affiliations:** 1Department of Medicine, Division of Pulmonary, Critical Care and Sleep Medicine, University of California Los Angeles, Los Angeles, CA 90095, USA; lyhsu@mednet.ucla.edu (L.H.); amaller@mednet.ucla.edu (A.M.); bcwalsh@mednet.ucla.edu (B.C.W.); djun@mednet.ucla.edu (D.J.); 2Barlow Respiratory Hospital, Los Angeles, CA 90026, USA; 3Department of Medicine, Division of Palliative Care Medicine, University of California Los Angeles, Los Angeles, CA 90095, USA; 4Department of Anesthesiology, András Jósa County Hospital, 4400 Nyíregyháza, Hungary; dr.szucs.attila@szszbmk.hu; 5Department of Internal Medicine, Division of Pulmonary and Critical Care Medicine, National Taiwan University Hospital, Taipei 100, Taiwan; jsjerng@ntu.edu.tw; 6Pulmonary, Critical Care and Sleep Section, West Los Angeles VA Medical Center, Los Angeles, CA 90073, USA

**Keywords:** prolonged mechanical ventilation, tracheostomy, specialized weaning units

## Abstract

Patients requiring mechanical ventilation (MV) beyond 21 days, usually referred to as prolonged MV, represent a unique group with significant medical needs and a generally poor prognosis. Research suggests that approximately 10% of all MV patients will need prolonged ventilatory care, and that number will continue to rise. Although we have extensive knowledge of MV in the acute care setting, less is known about care in the post-ICU setting. More than 50% of patients who were deemed unweanable in the ICU will be liberated from MV in the post-acute setting. Prolonged MV also presents a challenge in care for medically complex, elderly, socioeconomically disadvantaged and marginalized individuals, usually at the end of their life. Patients and their families often rely on ventilator weaning facilities and skilled nursing homes for the continuation of care, but home ventilation is becoming more common. The focus of this review is to discuss recent advances in the weaning strategies in prolonged MV, present their outcomes and provide insight into the complexity of care.

## 1. Introduction

Invasive positive pressure mechanical ventilation (MV) is a life-saving intervention commonly used in emergency rooms and in the intensive care unit (ICU) worldwide. It is estimated that 2–6% of all hospitalizations internationally result in the use of ventilatory support [1,2]. While we have gained significant scientific knowledge about the initiation of MV [3], including its benefits and risks [4], less is known about the cessation of MV. In seminal studies, Brochard et al. [5] and Esteban et al. [6] showed that repeated trials of weaning from MV are often needed for ventilator liberation. It has also been established that a shorter MV time leads to fewer complications in the ICU [7]. Unfortunately, approximately 15% of intubated patients require MV beyond 7 days [8], and 4.6% require it at 60 days [9]. International data show that patients who require prolonged MV beyond 21 days are generally considered to have a poor prognosis, with a 2-year mortality approaching 25% [3,10]. Despite these poor outcomes, the number of patients receiving prolonged MV is increasing and resulting in significant healthcare costs [11]. On the contrary, successful ventilator liberation after prolonged MV improved survival and decreased healthcare utilization [12,13]. In this review, we will summarize recent advances made in the ventilator weaning of patients receiving prolonged MV.

## 2. Prolonged MV, Ventilator Dependence and Long-Term Ventilation

Per Center for Medicare Services (CMS), prolonged MV is defined as at least 21 consecutive days of MV for at least 6 h daily [14]. Ventilator dependence is less well defined but generally refers to patients who are unable to liberate from MV in the ICU [15]. Long-term ventilation may describe two separate groups of patients in the scientific literature: (1) those with a prolonged MV need regardless of the mode of ventilation, invasive or non-invasive [16], and (2) MV patients who are no longer weaning candidates [17]. It is estimated that 6 to 10% of patients who are initially placed on MV end up on prolonged MV [12,18,19]. Long-term dependence on MV varies greatly depending upon the pathology of the patient’s respiratory failure—most commonly, chronic respiratory diseases, neuromuscular disorders and spinal cord injuries [16]. Furthermore, many patients receive tracheostomy for prolonged MV, who are often kept long-term. However, it remains unknown how the clinical outcomes of patients living with tracheostomies without MV compare to those who are continuously dependent on MV [16]. While prolonged MV, ventilator dependence and long-term ventilation are often used interchangeably, they may refer to different groups of patients. We use prolonged MV in this review to describe patients who require invasive MV after ICU discharge but remain eligible for ventilator weaning.

## 3. The Role of Tracheostomy in Prolonged MV

While surgical tracheostomy has been practiced for centuries, the advent of percutaneous tracheostomy in the 1980s was instrumental in popularizing its application today [20,21]. It is estimated that 8% of all MV patients undergo tracheostomy [22]. The age-adjusted rates of tracheostomy among all patients with MV increased from 16.7 to 34.3 cases per 100,000 adults from 1993 to 2012 in the USA [23]. Percutaneous tracheostomies are able to be performed outside the operating room, provide long-term safe airway access, reduce sedation needs, lower the frequency of ventilator-associated pneumonia and shorten the MV and hospital length of stay [24,25]. For patients requiring prolonged MV, tracheostomies have allowed for earlier acute care hospital discharges to outside facilities [23]. However, the exact timing of tracheostomy remains unknown. A recent study showed that tracheostomies performed within 7 days of ICU admission did not provide a mortality benefit when compared to those performed later [24]. 

## 4. Factors Contributing to Prolonged MV

The International Consensus Conference (ICC) described the disease pathology resulting in MV and published recommendations on how to assess patient readiness for ventilator liberation after critical illness [8]. These guidelines focus on the resolution of respiratory and hemodynamical instability. In patients with a prolonged MV need, chronic underlying medical conditions and sequelae of acute illness also significantly contribute to weaning difficulty. The most common reasons for inability to wean are: complications of acute respiratory failure, chronic respiratory diseases, uncontrolled diabetes mellitus, neuromuscular diseases, sequelae of cerebrovascular accidents, spinal cord injuries, malnutrition, morbid obesity and chronic heart failure. Optimizing these conditions takes time, but adequate management can significantly contribute to the success of ventilator liberation. 

## 5. Acute Respiratory Failure and Prolonged MV

Acute respiratory failure is the leading cause of MV initiation [22], and 67% of all MV patients meet the criteria for acute respiratory distress syndrome (ARDS) during their ICU stay [26]. ARDS is a severe form of acute lung injury causing acute respiratory failure and leading to high mortality [26]. Less is known about survivors of ARDS, but the work of Gajic et al. suggests that two-thirds of patients with persistent lung injury will require MV beyond 2 weeks [27]. Recently, molecular and clinical phenotyping provided evidence that ARDS patients can be divided into hyperinflammatory and non-hyperinflammatory groups [28]. The hyperinflammatory subphenotype is marked by increased plasma inflammatory biomarkers, severe shock and profound metabolic acidosis leading to longer MV and higher mortality when compared to those with non-hyperinflammatory responses [29]. Data also suggest that patients with a certain genetic makeup are more prone to develop ARDS and may have a less favorable response to MV [30]. Further research is needed to identify groups of ARDS patients that may require prolonged MV.

## 6. Spontaneous Breathing Trials

The assessment of a patient’s readiness to breathe with little to no assisted ventilation, known as a spontaneous breathing trial (SBT), is generally considered to be the first major step towards liberation from MV. A successful trial signals that the patient has a sufficient ability for spontaneous breathing. Assessment for spontaneous breathing has been recommended as early as possible in hemodynamically stable patients, as patients who can disconnect from the ventilator earlier have better clinical outcomes [8,31]. The optimal mode and length of SBTs have been debated. SBTs were traditionally performed by directly disconnecting from the MV and applying oxygenated air (T-piece trial) [5]. More recently, pressure support ventilation (PSV) trials have also been used [32]. While some showed no significant difference in the clinically meaningful outcomes between T-piece versus PSV trials [33], others have favored the latter [34]. In patients with chronic obstructive lung disease (COPD) receiving MV for more than 15 days, PSV versus T-piece trials yielded the same weaning success [35]. Due to a lack of clear evidence on how SBTs should be performed, international guidelines do not recommend a specific SBT method [8,36]. In PSV trials, less than 8 cmH_2_O of pressure support (PS) has been applied to low (0 to 5 cmH_2_O) positive end expiratory pressure (PEEP) to compensate for the endotracheal tube length and diameter. These trials can be routinely performed by built-in protocols in modern mechanical ventilators (Automatic Tube Compensation, ATC trial). In a comparative analysis, the two forms of SBTs showed similar weaning success [37]. The scientific literature also suggests that a shorter and less demanding 30 min SBT is sufficient to access MV liberation in the general ICU population [34,38]. However, it is less well known how to apply this knowledge to patients with prolonged MV, because many have undergone tracheostomy and have significant generalized weakness [8]. Jubran et al. studied tracheostomized patients requiring MV and found that a 12 h SBT is useful for differentiating between those patients who require continued MV and those who can be safely liberated [39]. Extended SBT times between 1 and 12 h have been proposed by multiple research groups for tracheostomized patients [40,41]. To help evaluate the readiness for SBTs in prolonged MV, researchers have evaluated the benefit of the rapid shallow breathing index (RSBI), calculated by dividing the respiratory rate (breath/minute) by the average tidal volume ventilation in liters. This technique was initially developed to help with the assessment for endotracheal tube extubation, but subsequent applications in the tracheostomized population are also validated. The research of Chao and Scheinhorn showed that an RSBI less than 80 signals SBT readiness [41], and Yang et al. used the RSBI to predict the trajectory of the weaning outcome in prolonged MV [42].

## 7. Ventilator Weaning and Liberation

Ventilator weaning refers to the process of liberation from MV. To unify nomenclature around ventilator weaning and liberation, the ICC identified three distinct group of patients based on the success of SBTs [8]. Weaning success was defined as passing the SBT and the cessation of MV for more than 48 h after extubation. Weaning failure was described as a failed SBT, reintubation and a return to MV or death within 48 h. Group 1 patients (simple weaning) passed their SBT on the first attempt. Group 2 patients (difficult weaning) required up to three SBTs within 7 days to be successfully weaned from MV. Group 3 patients (prolonged weaning) needed more than three SBTs or greater than 7 days for successful weaning (Table 1). The ICC classification identifies patients who will easily wean (Group 1 and 2) but did not discuss in detail the heterogeneous Group 3 patients. To help with this knowledge gap, the Weaning Outcomes according to a New Definition (WIND) classification was created [9]. WIND grouped patients based on the time needed for the first separation attempt from MV with or without SBTs. Group 1 (short weaning) patients separated from MV or died within 24 h. Group 2 (difficult weaning) patients separated from MV between 1 and 7 days, and Group 3 (prolonged weaning) patients separated from MV or died beyond 7 days. Group 3 was further divided into Group 3a for those who eventually separated from the ventilator and Group 3b for those who did not. An additional group, Group “no weaning”, was also created to describe those who never had a separation event from the ventilator. The authors of WIND also created a new nomenclature, in which ventilator weaning was described as separation from MV and weaning success was described as patients not needing MV for at least 7 days. They have also acknowledged that patients with tracheostomy may have a different weaning process and described the ventilator weaning attempt as the first successful 24 h without MV. This helped to further classify patients with complicated weans, but the exact approach to weaning in prolonged MV remains unknown. In Table 1, we compared the nomenclature of ICC and WIND classifications.

## 8. Modes of Ventilator Weaning in Prolonged MV

Given the heterogeneity of prolonged MV patients, there is currently no consensus on MV weaning modalities. In prolonged MV, tracheostomies are almost always used to facilitate weaning. The benefits of a protocolized approach to MV weaning was shown in a large metanalysis [43], while others highlighted that frequent assessments lead to shorter durations of MV and more successful outcomes [44]. Early studies also showed that protocolized, therapist-lead weaning resulted in improved MV liberation because it limited observer bias [35,45]. In Figure 1, we summarize protocolized approaches to prolonged MV weaning with tracheostomy. There is a general consensus that if a patient is able to successfully undergo an SBT, then unassisted breathing trials can be conducted under close supervision. While the minimum time period from MV cessation to the determination of ventilator liberation is not clear, the WIND criteria suggest that 7 days of unassisted breathing assures safe permanent disconnect [9]. The mode of unassisted breathing trials may vary center by center, but they all use a zero PEEP challenge. Jubran et al. recommended up to 24 h of unassisted breathing trials through a humidified oxygen delivery device connected to the tracheostomy tube (tracheostomy collar, TC) for 5 days [39]. Wu et al. used continued ATC trials for 3 days [40]. If a patient passes the SBT but fails at unassisted breathing within 12 h, a daily repeated challenge with unassisted breathing (TC or ATC trials) can be applied until the patient is able to tolerate at least 24 h of TC [39]. Other protocols by Surani et al. [46] and Scheinhorn et al. [45] call for incremental daily TC trials for up to 7 days. Patients who have a stable respiratory and hemodynamical status but fail SBTs may still be considered for weaning. The gradual decrease in PS with PSV has been preferred as a weaning mode in this chronically ill group of patients [8], largely based on initial studies performed in the ICU [5,6]. PSV weaning is performed by the gradual decrease in PS by 2 cmH_2_O from 20 cmH_2_O to 8 cmH_2_O and by maintaining the respiratory rate below 30/minute. In PSV weaning, PSV is alternated with assist control for 6–12 h daily to prevent respiratory muscle fatigue. More recently, Jubran et al. compared PSV weaning with daily TC trials. They found that in patients who failed the SBT but tolerated more than 12 h of TC time at the first assessment, continued TC trials resulted in a greater weaning success and a shorter weaning time than PSV weaning (weaning success 71% vs. 38.5% and weaning time 9 days vs. 20 days, respectively) [39]. For those who could not tolerate TC for at least 12 h, the weaning success and time to wean was similar between TC and PSV weaning (weaning success 49% vs. 46% and weaning time 16 days vs. 19 days, respectively) [39]. Of note, in this study, weaning success was declared if the patient was able to tolerate 5 days of consecutive unassisted breathing. These data suggest that in the chronically critically ill with an ongoing poor respiratory status, individualized weaning approaches may be needed. The Therapist-Implemented Patient-Specific (TIPS) weaning is a hybrid mode utilizing Synchronized Intermittent Mandatory Ventilation (SIMV), PSV and TC weaning sequentially [45]. While SIMV fell out of favor in ventilator weaning because of the poor performance [6]; its benefit in a complex weaning program is not known. With the flexibility of easy transition among various weaning modes, TIPS allows for adjustments depending on the patient’s daily need without the complete restart of the weaning process. Further investigation is needed to identify groups of patients who would benefit from individualized approaches after previous weaning failures; examples include the use of diaphragmatic pacemakers in bilateral diaphragmatic paralysis patients with spinal cord injuries [47], and, more recently, catheter-directed pacing of the diaphragm in chronic critical illness has also been tried [48,49].

## 9. The Role of Non-Invasive Ventilation in Prolonged MV Weaning

Non-invasive ventilation (NIV) via tight fitting face masks has been used in the ICU to help with the ventilator weaning of patients who failed the initial SBT [50] and also to prevent postextubation respiratory failure [51]. However, its role in prolonged MV weaning is less explored. In tracheostomized patients, Sancho et al. successfully used NIV via nose–mouth masks to reduce the time of invasive ventilation [52]. In a carefully selected patient population of tracheostomized patients who were able to complete at least 8 h of spontaneous breathing with capped trachesotomy tubes, Ceriana et al. showed that NIV can be used as an alternative measure to invasive ventilation [53]. Patients who live with chronic hypercapnia (described as a daytime partial arterial carbon dioxide (paCO_2_) pressure greater than 45 mmHg [8]), can use NIV for extended periods of time or even continuously as an alternative to invasive ventilation via tracheostomy. Studies with obstructive sleep apnea [54,55], obesity hypoventilation syndrome [56], neuromuscular diseases [57] and COPD patients [55,58] show the feasibility of chronic ventilation with NIV, but compliance and the technical challenges of mask wearing limit their general applicability outside of the ICU. While detailing the various forms of long-term NIV approaches is beyond the scope of this review, we would like to acknowledge the growing use of NIV in chronic respiratory failure [59,60]. In addition to NIV via face masks, high flow nasal cannulas (HFNC) have recently been applied to patients with chronic respiratory failure to prevent and to replace invasive ventilation [61]. These cannulas are capable of providing up to 100% of oxygen with a high flow rate and a low level of PEEP, which in turn results in the flow-dependent reduction in carbon dioxide (CO_2_) in the airways [62]. While primarily used in the ICU, HFNC have shown promise in reducing hypercapnia and facilitating ventilation, making them an intriguing new tool in treating chronic stable hypercarbic respiratory failure outside of acute care settings [60,63], including at home [64].

## 10. Use of Speaking Valves with MV and Weaning

One of the most significant challenges of MV is the loss of the upper airway resulting in impaired swallowing and secretion management. This may lead to aspirations, infections as well as psychological distress to patients from losing a vital method of communication [65]. These issues become more apparent with prolonged MV due to the atrophy of laryngeal muscles and the impairment of vocal-cord closure reflexes. To enable upper airway rehabilitation, one-way or speaking valves (SV) were developed. SV allows for easy air entry through the opening of a valve during inspiration that closes during expiration. The expiratory air flow is redirected to the vocal cords, which in turn allows for vocalization in MV-dependent patients [66]. The advantages provided by the restoration of laryngo-pharyngeal air flow and subglottic pressure include: improved secretions, a return of coughs and glottis closing reflexes, breathing–swallowing coordination through protective expiration after swallowing and an improved quality of life with verbal communication [65,67]. Its applicability in ventilator-dependent patients has been demonstrated, and its safety has been shown within 24 h of percutaneous tracheostomy placement [68]. In terms of physiologic effects on ventilator weaning, there appears to be potential benefits in lung recruitment. Studies by Sutt et al. have shown that SV in tracheostomized MV patients results in increased end-expiratory lung impedance (indicative of alveolar recruitment), better respiratory mechanics and subsequent improvements in tidal volumes [69]. Despite the numerous advantages of SV, its role in MV weaning is less well defined. While SV can be safely applied in the ICU, secondary outcomes looking at the duration of MV, time to tracheostomy decannulation and ICU and/or hospital stay have not differed with the early application of SV when compared to the standard use in tracheostomized patients [70]. Additionally, clinical practice has varied, in part due to complex physiologic processes across different respiratory conditions, practitioner unfamiliarity and the lack of a protocolized approach. Multi-disciplinary collaborative research is needed to study its benefit in prolonged MV [71].

## 11. Tracheostomy Decannulation

Tracheostomy decannulation is considered the final step of recovery from respiratory failure. While there are significant data on the rate of tracheostomies [22,31], less is known about the success of decannulation. Decannulation rates may significantly vary between 64% and 86% by 1 year depending on the study population and geographical location [72,73]. There are multiple factors that contribute to the decision to decannulate. In addition to the maintenance of a patent’s airway and easy reconnection to MV, cuffed tracheostomy cannulas can circumvent upper airway obstruction and prevent aspiration. At the same time, chronic tracheostomy cannulas result in an abnormal airway anatomy leading to chronic cough, difficulty swallowing and aphonia [74]. There is approximately a 5% failure rate associated with decannulation [75]. Most practitioners use the general assessment of: (1) successful ventilator liberation for at least 5 days, (2) hemodynamic stability, (3) pCO_2_ less than 60 mmHg from blood gas analysis, (4) preserved cognition, (5) adequate swallow function and (6) no significant obstruction on the direct airway exam based on the recommendations of Ceriana et al. [53]. Of note, patients with a pCO_2_ value persistently higher than 60 mmHg may be able to proceed with decannulation in certain circumstances. O’Connor et al. suggested a trial of non-invasive positive pressure ventilation via a nasal or face mask with a capped tracheostomy tube in these patients, but pressure leaks via a large stoma may limit applicability [76]. Tracheostomy tubes also eliminate issues of sleep-disordered breathing, as the tube bypasses the upper airway. Upon decannulation, previously undiagnosed sleep-disordered breathing may be unmasked, which will require assessment with sleep testing [77]. More recently, an international survey suggested that age and underlying disease etiologies should also be considered when a decision is made not to decannulate a patient [78]. Airway-resistant devices including SV, downsizing the tracheostomy cannula, direct finger occlusion or capping the tracheostomy can be used as ancillary measures to assess for decannulation [74,79,80]. A practical approach to decannulation assessment is shown in Figure 1.

## 12. The Role of Complex Rehabilitation 

Besides medical care, skilled nursing, dedicated respiratory therapy, physiotherapy, speech and language pathology, occupational therapy, psychological care, wound care, nutritional support, care coordination and family support all contribute to the success of weaning in prolonged MV patients. While discussing the role of all these factors is beyond the scope of this review, we would like to acknowledge that a collaborative patient and center-specific approach is necessary for the successful rehabilitation of patients with a prolonged ventilation need [81].

## 13. Location of Ventilator Weaning

MV has originally been performed for a short period of time in the ICU with hopes of a quick recovery. By the 1980s, prolonged MV became increasingly common, resulting in the establishment of specialized weaning units (SWU) located in hospitals or as free-standing units [14]. In the United States, SWU is often combined with other forms of complex rehabilitation in Long-Term Acute Care Hospitals (LTACH). In Table 2, we summarized outcomes of SWUs published after ICC weaning recommendations became available in 2007. These studies show that the weaning success is generally higher than 50% but will vary based on the study population. Weaning units with frequent readmissions and higher levels of acuity may have poorer outcomes reflected by less weaning success, longer hospital lengths of stay or higher inpatient mortalities. Recently, we and others have demonstrated that patients recovering from severe COVID pneumonia with tracheostomies had better weaning outcomes than the general SWU population because they had less co-morbid conditions [82,83]. Because SWUs care for a large volume of patients and run successful weaning programs, they have reduced costs compared to acute care hospitals and ICUs [13,84]. However, the overall value of care in SWUs and LTACHs has been questioned due to the long care time and high mortality [10,85]. Herer et al. showed that SWUs may improve weaning success and shorten hospital stays but do not affect 1-year mortality when compared to weaning in acute care hospitals [86]. Unfortunately, outcomes are especially poor for those who cannot be liberated from MV in SWU and are transferred to nursing homes or subacute care units. The 1-year survival rate in this population is approximately 30% [87]. Home ventilation has been an emerging alternative for these patients, but the standards of home ventilator care have not been established [88]. Recently, Jacobs et al. argued that home ventilation programs can be as successful as ventilator care in the LTACH [89]. With the development of remote ventilator manipulation, home ventilator weaning may become a reality [90]. Home ventilation programs are especially beneficial for patients living with progressive neuromuscular diseases like Duchenne muscular dystrophy [91] and amyotrophic lateral sclerosis (ALS) [92], where tracheostomy and invasive MV is needed for survival and to maintain quality of life [93].

## 14. Goal-Concordant Care for Patients with Prolonged MV

Due to the poor outcomes in prolonged MV, the continued re-evaluation of patient preferences remains a significant part of compassionate care. While initial conversations soliciting treatment preferences for long-term respiratory support have become the standard of care in the ICU [97], these discussions can be significantly more challenging when patients are in a post-acute setting. While these patients have all opted for a prolonged trial of medical therapy with the goal of ventilator liberation, many are older than 55 years and live with chronic medical problems, which elevates their risk for unfavorable outcomes [10]. Some patients who are not achieving their rehabilitative goals may have a transition in their treatment preferences to comfort care and palliative liberation from the ventilator. Such conversations occur infrequently. Only 21.5% of patients discussed treatment preferences with a physician in European respiratory intermediate care units [98], and only 36.5% of LTACHs in the United States have access to a palliative care program [99]. Even though healthcare professionals agree on the generally poor outcomes of prolonged MV, goal-concordant palliative liberation from MV has been limited by the sparse availability of palliative programs, family perceptions, religious beliefs, geographical locations, healthcare laws and local policies [100,101,102]. Further research on the optimal integration of palliative care in prolonged MV is highly desired. 

## 15. Conclusions

The need for prolonged MV is increasing worldwide. A significant amount of research has evaluated the benefits and potential harms of continued ventilator care, and weaning Novel protocols enabled weaning success even beyond the ICU, but further research is needed to identify those who would benefit from ongoing weaning attempts. Unifying definitions and approaches to ventilator weaning in this patient population will likely result in improved care. 

## Figures and Tables

**Figure 1 jcm-13-01909-f001:**
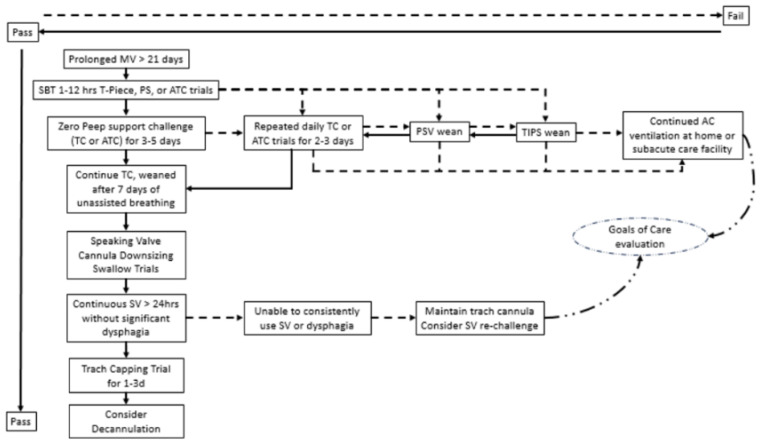
Mechanical ventilator weaning strategies in prolonged mechanical ventilation. Prolonged mechanical ventilation (MV) is defined by at least 6 h of daily ventilation beyond 21 days. Weaning ability should be assessed by spontaneous breathing trials (SBTs) as long as the patient’s hemodynamical and respiratory status is stable. SBTs can be performed by unassisted breathing via a T-piece with oxygen blowby, low pressure support (PS) or automatic tube compensation (ATC) support below 8 cmH_2_O for up to 12 h. If a patient passes the SBT, the patient is usually challenged with zero positive end expiratory pressure (PEEP) using unassisted breathing through a humidified oxygen delivery device connected to the tracheostomy tube (tracheostomy collar, TC) or ATC with zero PEEP for up to 3 to 5 days. A patient is considered weaned from the ventilator after 7 days of unassisted breathing. If a patient fails the zero PEEP challenge but remains in stable condition, they can be reassessed with repeated TC or zero PEEP ATC trials. For patients who fail the initial SBT, multiple approaches to weaning have been described: (1) TC weaning. In this mode, patients are challenged with unassisted breathing for up to 12 h, sometimes with a daily extension of TC time, and rested on assist control ventilation. (2) Pressure support (PS) ventilation (PSV) weaning. In PSV, patients are challenged daily with a sequentially reduced PS from 20 cmH_2_O to 8 cmH_2_O for up to 12 h. Patients are rested on assist control ventilation. (3) The Therapist-Implemented Patient-Specific (TIPS) weaning protocol is a combination of the Synchronized Intermittent Mandatory Ventilation (SIMV) mode, continued by PSV weaning and followed by TC weaning in a stepwise fashion. The PSV and TIPS modes may be preferred in patients with a marginal performance status. Following ventilator weaning, patients who can tolerate speaking valves (SV) and tracheostomy cannula downsizing and are able to swallow can be considered for tracheostomy capping. Patients who can continuously wear tracheostomy caps for 1 to 3 days can be safely decannulated. Tracheostomy cannulas are kept long-term for patients who failed multiple weaning attempts or cannot tolerate capping trials. Patients who fail weaning attempts transition to the home or subacute care level with the continuation of MV. Multiple failed weaning attempts necessitate the re-evaluation of goals of care. Abbreviations: MV = mechanical ventilation, SBT = spontaneous breathing trial, pass = pass SBT, fail = failed SBT, PS = pressure support, ATC = automatic tube compensation, PEEP = positive end expiratory pressure, TC = tracheostomy collar, PSV = pressure support ventilation, TIPS = Therapist-Implemented Patient-Specific weaning, SV = speaking valve, hrs = hours, d=days.

**Table 1 jcm-13-01909-t001:** Classification of ventilator weaning.

Classification	ICC	WIND
Group 1	Simple weaning: successful extubation after one SBTs	Short weaning: successful separation from MV or death within 24 h
Group 2	Difficult weaning: successful extubation after up to three SBTs in less than 7 days	Difficult weaning: successful separation from MV or death in 1 to 7 days
Group 3	Prolonged weaning: successful extubation after more than three SBTs or more than 7 days	Prolonged weaning: unsuccessful separation 7 days after the first attempt. Subgroup A: eventually separated from MV; B: not separated from MV
Group “no weaning”	-	No separation attempt from MV

ICC = International Consensus Conference [8], WIND = Weaning Outcome According to a New Definition [9], MV = mechanical ventilation.

**Table 2 jcm-13-01909-t002:** Ventilator weaning trials of tracheostomized patients in specialized weaning units.

Author	Year	Country	N	Weaned (%)	Wean Time (Day)	LOS (Day)	Inpatient Mortality (%)
Bonnici [94]	2016	United Kingdom	168	61	19	31	14.5
Bornitz [95]	2020	Germany	65	79	-	21	1.6
Dolinay [82]	2022	USA	165	70	-	24	9.5
Ghiani [96]	2020	Germany	263	47.9	22	52	14.4
Herer [86] Cohort 2	2020	France	103	43.8	21	29	7.3
Jubran [39] PSV study arm	2013	USA	152	45	19	41	15
Jubran [39] TC study arm	2013	USA	160	53	15	42	10
Saad [83]	2022	USA	158	70.9	11	41	9.6
Scheinhorn [18]	2007	USA	1419	54	15	40	25
Surani [46]	2022	USA	111	89	8	-	21
Wu [40] ATC study arm	2023	Taiwan	157	62	-	-	13
Wu [40] TC study arm	2023	Taiwan	246	71	-	-	18

N = number of patients, weaned = patients considered liberated from mechanical ventilation in percent, wean time = median time to liberation in days, LOS = average weaning unit length of stay in days, inpatient mortality = % mortality at weaning units, PSV = pressure support ventilation, TC = trach collar, ATC = automated tube compensation.

## Data Availability

Not applicable.

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
