# Peer review of "Ventilator Weaning in Prolonged Mechanical Ventilation—A Narrative Review"

_jcm, 2024, doi:10.3390/jcm13071909_

Round 1
Reviewer 1 Report
Comments and Suggestions for Authors
Thank you for the opportunity to review the manuscript entitled "Prolonged mechanical ventilation-a narrative review"
This narrative review is very interesting and well written.
I have one comment that I believe could improve the manuscript.
The authors should include one or two paragraphs to discuss subphenotyping of critically ill/mechanically ventilated patients. Recent literature has shown that inflammatory subphenotypes in patients with ARDS (Sinha et al. PMID: 34253679, Calfee et al. PMID: 24853585), at risk of ARDS (Redaelli et al. PMID: 37906258) and septic patients (Since et al. PMID: 37633303) are markers of mortality, decreased ventilator-free days, longer ICU and H stay etc. These subohenotypes might be useful in identifying patients at risk of prolonged mechanical ventilation, clinical deterioration, weaning failure. The authors should also discuss the limitations of these subphenotypes: clinical applicability, generalisability, reproducibility.
Author Response
The authors would like to thank the reviewer for the useful comments. We revised the title of our manuscript to emphasize that the focus of this review article is ventilator weaning in prolonged mechanical ventilation.
Please see our point-by-point response below.
The authors should include one or two paragraphs to discuss subphenotyping of critically ill/mechanically ventilated patients. Recent literature has shown that inflammatory subphenotypes in patients with ARDS (Sinha et al. PMID: 34253679, Calfee et al. PMID: 24853585), at risk of ARDS (Redaelli et al. PMID: 37906258) and septic patients (Since et al. PMID: 37633303) are markers of mortality, decreased ventilator-free days, longer ICU and H stay etc. These subohenotypes might be useful in identifying patients at risk of prolonged mechanical ventilation, clinical deterioration, weaning failure. The authors should also discuss the limitations of these subphenotypes: clinical applicability, generalisability, reproducibility.
We agree with the reviewer that acute respiratory failure and specifically acute respiratory distress syndrome (ARDS) is a major cause of mechanical ventilation. Following the reviewer recommendation, we included a new section “Acute respiratory failure and prolonged MV” to discuss data linking ARDS to mechanical ventilation (MV). We also cited work describing the hyperinflammatory and non-hyperinflammatory phenotypes in ARDS, which may lead in a differential response to MV. While ARDS subphenotyping does not currently alter MV techniques or guide ventilator weaning, we agree with the reviewer that it will be an important consideration as new evidences become available.
We hope our revised manuscript will merit your acceptance.
Sincerely,
Tamas Dolinay M.D., Ph.D.
Corresponding author
Reviewer 2 Report
Comments and Suggestions for Authors
The authors submit here a review on prolonged mechanical ventilation. This review seems me well-written and updated.
My remarks are as follows :
-The main aspects of prolonged mechanical ventilation are here described. However the word long-term MV appears only in the conclusion, without previous mention or definition. To my opinion the authors need to define long-term MV, a distinct entity from prolonged MV, in a dedicated paragraph with specific data.
-Page 2 line 77 : skin ulcers are a frequent complication in patients submitted to prolonged mechanical ventilation but not a reason of inability to wean. Please correct.
-Page 2 paragraph Spontaneous breathing trials. Some statements seem me wrong. The ICC 2007 (reference 9) did’nt recommend or favor PSV over T-piece. The study by Thille (ref 28) did’nt show superiority of one method vs the other (P-piece vs PSV). Please correct
-Page 4 line 154 : please detail the term trach collar which could not be familiar for the reader : as unassisted breathing through an oxygen-delivery device connected to a tracheostomy tube.
-Page 7 Location of ventilator weaning : plese discuss the study by Herer B Respiratory Care 2020 which showed that the implementation of a respiratory unit dedicated to the weaning of prolonged MV patients improved the rate of weaning but not 1-year survival.
-Conclusion line 327 : 30-40% of patients require long-term MV. This statement seems me questionable and is clearly confusing. In the text the rate of 10 % of prolonged MV was cited. Long-term MV was not defined in the manuscript nor the prevalence mentioned. Long-term MV ususally refers to patients needing definitive ventilatory assistance. Please clarify.
Author Response
The authors would like to thank the reviewer for the useful comments. We revised the title of our manuscript to emphasize that the focus of this review article is ventilator weaning in prolonged mechanical ventilation.
Please see our point-by-point response below.
- The main aspects of prolonged mechanical ventilation are here described. However the word long-term MV appears only in the conclusion, without previous mention or definition. To my opinion the authors need to define long-term MV, a distinct entity from prolonged MV, in a dedicated paragraph with specific data.
Thank you for this important comment. In the revised version of the manuscript we discuss the terms prolonged MV, ventilator dependence and long-term ventilation. To this end, we added a “Prolonged MV, ventilator dependence and long-term ventilation” section. Specifically, long-term ventilation has been used as an umbrella term for patients who needed prolonged ventilation with any type of ventilator support, invasive or non-invasive, but can also refer to patients not been able to liberate from the ventilator. In the revised version, we discuss these complex terminology and clarify that the focus of our review is ventilator weaning in prolonged mechanical ventilation.
- Page 2 line 77 : skin ulcers are a frequent complication in patients submitted to prolonged mechanical ventilation but not a reason of inability to wean. Please correct.
We agree with the reviewer that skin ulcers are primarily the result of prolonged MV and we removed it from the text.
3.Page 2 paragraph Spontaneous breathing trials. Some statements seem me wrong. The ICC 2007 (reference 9) did’nt recommend or favor PSV over T-piece. The study by Thille (ref 28) did’nt show superiority of one method vs the other (P-piece vs PSV). Please correct
We agree with the reviewer that current guidelines do not prefer one form of SBT over another. We have revised that text and added new references to show that both PSV and T-piece SBT are used.
- Page 4 line 154: please detail the term trach collar which could not be familiar for the reader : as unassisted breathing through an oxygen-delivery device connected to a tracheostomy tube.
We amended our text to explain the trach collar device, based on the reviewer’s guidance.
- Page 7 Location of ventilator weaning: please discuss the study by Herer B Respiratory Care 2020 which showed that the implementation of a respiratory unit dedicated to the weaning of prolonged MV patients improved the rate of weaning but not 1-year survival.
The authors would like to thank the reviewer for this reference. In the revised text, we discuss the lack of mortality benefit of specialized weaning units compared to in-hospital weaning and we also included data from this paper in our revised Table 2.
- Conclusion line 327 : 30-40% of patients require long-term MV. This statement seems me questionable and is clearly confusing. In the text the rate of 10 % of prolonged MV was cited. Long-term MV was not defined in the manuscript nor the prevalence mentioned. Long-term MV usually refers to patients needing definitive ventilatory assistance. Please clarify.
We discuss the relevant terminologies in the revised “Prolonged MV, ventilator dependence and long-term ventilation” section. We agree with the reviewer that using long-term ventilation in the Conclusion can lead to misunderstanding and we removed it. In the revised Conclusion, we discuss that 30-40% of patients who receive prolonged MV will not liberate from the ventilator and further research is needed to identify those who would benefit from ongoing weaning attempts.
We hope our revised manuscript will merit your acceptance.
Sincerely,
Tamas Dolinay M.D., Ph.D.
Corresponding author
Reviewer 3 Report
Comments and Suggestions for Authors
Thanks for the opportunity to review this original article by dr Tamás Dolinay et al. titled: Prolonged mechanical ventilation-a narrative review
The study is interesting and focuses on reviewing the current literature on the topic of prolonged invasive ventilation. I congratulate to the authors for the interesting topic.
I suggest few comments that I would like to be better addressed in the manuscript
1. The authors have not mentioned the use of other respiratory supports in helping weaning the patient off invasive mechanical ventilation completely or just as a bridge to it or during decannulation.
For instance some chronic respiratory diseases such as COPD and Neuromuscolar diseases may be weaned off onto noninvasive respiratory (NIV) support. Please expand considering that the use of the correct mask may play a pivotal role in the weaning process. Please see ref doi: 10.1080/17476348.2022.2121706
2. Other respiratory supports may be used in support on the weaning process like for example HFNC in COPD patients: please see ref: 10.1186/s13054-018-2107-9 , doi: 10.1016/j.jcrc.2010.06.003, doi: 10.1164/ajrccm.164.2.2008160. and expland the topic.
3. Line 271/2: this is true but it is also true that the combination of some or all of these facilities depending on what is available on the location may be successful in the complete weaning of the patients. Please revise.
Minor: Please do not start a sentence with an acronym. There is a number of misspelling errors and some phrases that need to be rephrased in the manuscript, please provide a mother tongue English revision and mention it in the acknowledgments.
Thanks again for this opportunity. I look forward to revise the next version of the manuscript.
Comments on the Quality of English Languageminor revisions needed
Author Response
The authors would like to thank the reviewer for the useful comments. We revised the title of our manuscript to emphasize that the focus of this review article is ventilator weaning in prolonged mechanical ventilation.
Please see our point-by-point response below.
- The authors have not mentioned the use of other respiratory supports in helping weaning the patient off invasive mechanical ventilation completely or just as a bridge to it or during decannulation. For instance some chronic respiratory diseases such as COPD and Neuromuscular diseases may be weaned off onto noninvasive respiratory (NIV) support. Please expand considering that the use of the correct mask may play a pivotal role in the weaning process. Please see ref doi: 10.1080/17476348.2022.2121706.
We would like to thank the reviewer for this constructive suggestion. To elaborate on the role of non-invasive ventilation (NIV) in prolonged mechanical ventilation (MV) weaning, we added a new section: “The role of non-invasive ventilation in prolonged MV weaning”. In this section we discuss:
1) The use of tight fitting masks to support the weaning of patients from MV who have a tracheostomy and are capable of some unassisted breathing. We show the results of 2 observational studies.
2) NIV to prevent or replace invasive mechanical ventilation in patients with chronic respiratory failure. Here, we mention diseases with specific needs for long-term NIV, including obstructive sleep apnea, obesity hypoventilation syndrome, neuromuscular diseases and COPD. Because the focus of our manuscript is ventilator weaning in prolonged invasive mechanical ventilation, for further details about long-term NIV, we referred to relevant articles including Pierucci et al. The right interface for the right patient in noninvasive ventilation: a systematic review. Expert Rev Respir Med 2022; 16: 931-944.
3) The role of high-flow nasal cannula (HFNC) in preventing and replacing invasive mechanical ventilation in patients with stable hypercarbic respiratory failure outside of the ICU.
- Other respiratory supports may be used in support on the weaning process like for example HFNC in COPD patients: please see ref: 10.1186/s13054-018-2107-9 , doi: 10.1016/j.jcrc.2010.06.003, doi: 10.1164/ajrccm.164.2.2008160. and explained the topic.
We discuss the use of HFNC in the “The role of non-invasive ventilation in prolonged MV weaning” section. Please see above. We cite relevant articles, including the one suggested by the reviewer: Di mussi et al. High-flow nasal cannula oxygen therapy decreases postextubation neuroventilatory drive and work of breathing in patients with chronic obstructive pulmonary disease. Critical Care 2018; 22:180.
Patients with progressive neuromuscular disease, including those with amyotrophic lateral sclerosis (ALS), often need life-long MV. Data suggest that long-term ventilation can improve quality of life in ALS. While home ventilation is less common in the USA than in Europe, trends suggest that the need for home ventilation will grow worldwide. We agree with the reviewer that this is an important topic and we discuss this in our revised “Location of weaning” section. Here we cite among other the articles, Vianello et al. Survival and quality of life after tracheostomy for acute respiratory failure in patients with amyotrophic lateral sclerosis J Crit Care 2011;26:e7-14. In addition to respiratory devices, we discuss the use of diaphragmatic pacemakers to facilitate weaning in patients with diaphragmatic weakness in the revised “Modes of ventilator weaning in prolonged MV” section.
Spontaneous breathing trials (SBT) are critical in determination of weanability in prolonged MV. We updated our “Spontaneous breathing trials” section to discuss specific data on COPD patients requiring more than 15 days of MV. In this patient population, SBT performed with pressure support ventilation (PSV) or with unassisted breathing via T-piece resulted in the same ventilator liberation success. We cite, Vitacca et al. Comparison of Two Methods for Weaning Patients with Chronic Obstructive Pulmonary Disease Requiring Mechanical Ventilation for More Than 15 Days Am J Respir Crit Care Med 2001;164:225–230.
- Line 271/2: this is true but it is also true that the combination of some or all of these facilities depending on what is available on the location may be successful in the complete weaning of the patients. Please revise.
We agree with the reviewer that a patient and center-specific approach is needed for the successful rehabilitation of patients with prolonged MV need. We rephrased our sentence accordingly.
- Minor: Please do not start a sentence with an acronym. There is a number of misspelling errors and some phrases that need to be rephrased in the manuscript, please provide a mother tongue English revision and mention it in the acknowledgments.
We revised our manuscript to correct grammatical errors.
We hope our revised manuscript will merit your acceptance.
Sincerely,
Tamas Dolinay M.D., Ph.D.
Corresponding author
Round 2
Reviewer 2 Report
Comments and Suggestions for Authors
The authors responded to my comments and their responses and corrections are mostly adequate.
However I still have a major concern regarding the conclusion. The sentence approximately 30-40% of prolonged MV patients do not liberate from the ventilator is highly questionable. This sentence is not supported by the main text, where this rate does’nt appear. The authors have to mention the rate of prolonged MV patients not weaned from MV in the main text, citing a reference in the literature if available, or to delete this sentence in the conclusion. Please correct.
Author Response
The authors would like to thank the reviewer for the repeat review. We omitted the data referenced by the reviewer to clarify our conclusion.
We hope that the revised manuscript will merit your acceptance.
Sincerely,
Tamas Dolinay MD. PhD.
corresponding author
Reviewer 3 Report
Comments and Suggestions for Authors
Thanks for the opportunity to review this manuscript again.
It has improved and I do not have any further comment.
Regards and good luck
Author Response
The author would like to thank the reviewer for the review and acceptance.
Sincerely,
Tamas Dolinay MD, PhD
corresponding author